# Osteogenesis Imperfecta: Search for Mutations in Patients from the Republic of Bashkortostan (Russia)

**DOI:** 10.3390/genes13010124

**Published:** 2022-01-10

**Authors:** Dina Nadyrshina, Aliya Zaripova, Anton Tyurin, Ildar Minniakhmetov, Ekaterina Zakharova, Rita Khusainova

**Affiliations:** 1Institute of Biochemistry and Genetics—Subdivision of the Ufa Federal Research Centre of the Russian Academy of Sciences, 450054 Ufa, Russia; a.ramilna@bk.ru (A.Z.); minniakhmetov@gmail.com (I.M.); ritakh@mail.ru (R.K.); 2Departament of Genetics and Fundamental Medicine, Bashkir State University, 450076 Ufa, Russia; 3Republican Medical Genetics Centre, 450076 Ufa, Russia; 4Internal Medicine Department, Bashkir State Medical University, 450008 Ufa, Russia; anton.bgmu@gmail.com; 5Research Centre for Medical Genetics, 115478 Moscow, Russia; labnbo@yandex.ru

**Keywords:** next-generation sequencing (NGS), I type of collagen, metabolic bone disease

## Abstract

Osteogenesis imperfecta (OI) is an inherited disease of bone characterized by increased bone fragility. Here, we report the results of the molecular architecture of osteogenesis imperfecta research in patients from Bashkortostan Republic, Russia. In total, 16 mutations in *COL1A1*, 11 mutations in *COL1A2*, and 1 mutation in *P3H1* and *IFIMT5* genes were found in isolated states; 11 of them were not previously reported in literature. We found mutations in *CLCN7*, *ALOX12B, PLEKHM1, ERCC4, ARSB, PTH1R*, and *TGFB1* that were not associated with OI pathogenesis in patients with increased bone fragility. Additionally, we found combined mutations (c.2869C>T, p. Gln957* in *COL1A1* and c.1197+5G>A in *COL1A2*; c.579delT, p. Gly194fs in *COL1A1* and c.1197+5G>A in *COL1A2*; c.2971G>C, p. Gly991Arg in *COL1A2* and c.212G>C, p.Ser71Thr in FGF23; c.-14C>T in *IFITM5* and c.1903C>T, p. Arg635* in *LAMB3*) in 4 patients with typical OI clinic phenotypes.

## 1. Introduction

Osteogenesis imperfecta (OI, Q78.0 according to ICD-10) is a rare genetic metabolic disease of the bone system with an autosomal dominant or a recessive type of inheritance. There are also X-linked forms and sporadic cases of this disease. The frequency of the disease in general varies from 1:15,000 to 1:20,000 [1].

The disease is characterized by bone fragility, skeletal deformity, short stature, blue sclera, progressive hearing loss, and dentin anomaly. According to the modern classification, the disease is divided into five types [2,3]. OI is a clinically and genetically heterogeneous hereditary connective-tissue disorder caused by the structural and quantitative changes in collagen as well as disorders associated with its post-translational modification, folding, and intracellular transport. To date, scientists have identified 21 genes that are responsible for the development of OI. Most patients with OI (80–90%) have autosomal dominant inheritance caused by mutations in the *COL1A1* or *COL1A2* genes encoding α-1 and α-2 chains of type I collagen [4]. Later, another gene was identified—*IFITM5*, mutations in which are responsible for the autosomal dominant type V of OI [5]. Mutations in this gene occur in 4–5% of patients. Less than 10% of patients with OI have recessive forms of inheritance caused by mutations in genes encoding proteins that are involved in the synthesis, transport, and post-translational modifications of collagen or factors associated with differentiation and mineralization of bone cells (*CRTAP, PPIB, BMP1, CCDC134, CREB3L1, FAM46A, FKBP10, P3H1, P4HB, PLOD2, PLS3, SEC24D, SERPINF1, SERPINH1, SP7, SPARC,* and *TMEM38B*) [1,3,6].

Currently, to find the molecular cause of a genetically heterogeneous disease such as OI, the most effective method is next-generation sequencing (NGS), a rapidly developing massive parallel sequencing technology that allows the analysis of multiple genes simultaneously.

The aim of the work is to search for pathogenic changes in target genes in patients from the Republic of Bashkortostan with osteogenesis imperfecta by NGS technology.

## 2. Materials and Methods

Patients. Our study involved 62 patients (mean age 24.6 ± 15.56) from 52 families living in the Bashkortostan Republic of Russia. All of them had clinical manifestations specific to osteogenesis imperfecta—multiple fractures, blue sclera, deformities, deafness, dentinogenesis imperfecta, joint hypermobility, and radiological data—and were included in NGS research. Type I of OI was determined in 23 patients (54.7%, 23/42), type III in 10 patients (23.8%, 10/42), type IV in 6 patients (14.3%, 6/42), and type V in 3 patients (7.2%, 3/42).

Preparatory stage. Whole venous blood of the patient is required for DNA diagnostics. The genomic DNA of the subjects was isolated from whole venous blood by phenol-chloroform extraction. The DNA concentration was measured using an Epoch-1 spectrophotometer (BioTek, Santa Clara, CA, USA) and Qubit (Thermo Fisher Scientific, Waltham, MA, USA). After measuring the DNA concentration, the samples were considered ready for next-generation sequencing (NGS).

NGS. In order to search for pathogenic variants that are likely to be the cause of the disease, several stages of research were conducted (Figure 1).

At the first stage, targeted NGS of type I collagen genes (*COL1A1*, *COL1A2*) was carried out according to the high frequency (85–90%) of mutations in this localization.

To analyze mutations in the *COL1A1* and *COL1A2* genes, a primer panel was developed for sequencing the complete gene sequence using targeted mass parallel sequencing (MPS). Enrichment of the genome regions of interest was carried out using PCR of long fragments using the BioMaster LR HS-PCR-Color (2×) kit (Biolabmix, Novosibirsk, Russia) with the production of fragments on a SureCycler 8800 amplifier (Agilent Technologies, Santa Clara, CA, USA). The presence of amplified products was checked by agarose gel electrophoresis. DNA libraries were prepared using the Nextera Flex kit (Illumina, San Diego, CA, USA) according to the manufacturer’s recommended protocol. MPS was performed on a MiSeq sequencer (Illumina, San Diego, CA, USA) using a MiSeq Reagent Kit V2 (Illumina, San Diego, CA, USA) sequencing kit according to the protocol recommended by the manufacturer. The data obtained as a result of the MPS underwent bioinformatic processing, which included the following key stages: assessing the quality of readings using the FastQC program, removing adapters and trimming the sequence using the Trimmomatic-0.36 package, aligning to target sequences using Bowtie2, converting to bam files, sorting them and converting them to VCF format using the Samtools set of utilities.

The study was carried out using the equipment and protocols of the “Medical Genomics” center (Tomsk, Russia) [7].

At the second stage, targeted NGS of 13 genes was carried out, the products of which are involved in collagen modifications in patients with no mutations in type I collagen genes.

Molecular genetic diagnostics of patients was carried out in the laboratory of the Department of Hereditary Endocrinopathies of the Federal State Budgetary Institution “National Medical Research Center of Endocrinology” of the Ministry of Health of Russia, Moscow. For DNA diagnostics, the author’s panel of primers for multiplex polymerase chain reaction (PCR) and sequencing using the Ion Ampliseq™ Custom DNA Panel technology (Life Technologies, Carlsbad, CA, USA) was used. The author’s panel “osteogenesis imperfecta” included 13 genes whose products are involved in collagen modifications. Library preparation and emulsion PCR were performed according to the manufacturer’s recommendations. At the preparatory stage (preparation of DNA libraries), the following was carried out: amplification of the studied regions of the genome, attaching adapters with 10 bar codes to them; purification of libraries; amplification of libraries on microparticles; and enrichment of microparticles containing DNA templates. Next, the DNA sequence was deciphered (sequencing on a semiconductor sequencer Personal Genome Machine (Ion Torrent, Life Technologies, Gilford, NH, USA). Bioinformatic processing of sequencing results was carried out using the Torrent Suite 4.2.1 software module (Ion Torrent, Life Technologies, USA) and the Annovar software package (version 2014Nov12) [8].

At the third stage, targeted NGS of 166 genes involved in bone metabolism of connective tissue in patients with no pathogenic changes at the previous stages of the study was carried out. This analysis was carried out at the Medical Genetic Research Center (Moscow) using an ION S5 (Thermo Fisher Scientific, Waltham, MA, USA) device, with a reading depth of at least ×110. This means that each investigated genome region is analyzed on average at least 110 times to avoid the influence of technical reading errors on the research results. For complex regions of the genome (for example, GC-rich regions), the average coverage may be lower.

At the fourth stage, targeted NGS of 664 genes involved in the metabolism of connective tissue in patients with no mutations in the genes of the target panels of the previous stages of the study was carried out.

Molecular genetic diagnostics of patients was carried out in the laboratory of molecular pathology “Genomed” (Moscow) using the MGISEQ-200 planform and kits (Beijing Genomics Institute, PRC). For molecular genetic analysis, the author’s panel “connective tissue diseases” was used. DNA analysis was carried out using new-generation sequencing technology using the pair-end reading method. The average coverage of the target sequencing sites in the studied genes was at least 70×.

Sequencing data processing was carried out using an automated algorithm that included an alignment of reads to the reference sequence of the human genome (hg19), post-processing of alignment, identification of variants, and filtering of variants by quality.

All identified mutations were confirmed by Sanger sequencing for patients and closer relatives using the Applied Biosystems 3500 genetic analyzer (Thermo Fisher Scientific Inc.) and the BigDye™ Terminator v3.1 cycle sequencing kit according to the manufacturer’s protocol (Applied Biosystems™ by Thermo Fisher Scientific).

Annotation of the identified variants according to the canonical transcript of each gene and their prioritization was performed, taking into account the ACMG recommendations version 3.0.

Standard nomenclature was used to describe nucleotide sequence variants [9].

Genomic coordinates were determined in accordance with the genomic assembly standard (GRCh38) [10]. The clinical significance and phenotypic manifestations of nucleotide sequence variants were determined using OI databases [6] and on the basis of the Exome Aggregation Consortium [11] as well as literature data.

To assess the functional significance of the identified changes in the nucleotide sequence in target genes, various databases and predictive programs were used. The search for the previously described variants was carried out in the databases of exome sequencing (Exome Aggregation Consortium; Exome Variant Server), genomic and targeted sequencing (1000 Genomes Project), single nucleotide variants (dbSNP) and structural variants (dbVar), and a specialized database on mutations and polymorphic variants for OI (Osteogenesis Imperfecta Variant Database). If a nucleotide variant was not previously described in the literature and is not presented in databases, or information about it is insufficient to decide on its significance, an analysis of the pathogenicity of the identified gene variants was carried out using the predictive programs SIFT (Sorting Intolerant From Tolerant), FATHMM (Functional Analysis through Hidden Markov Models), MutationAssessor, PolyPhen2 (Polymorphism Phenotyping v-2), Condel (Consensus Deleteriousness), MutationTaster, MutPred (Mutation Prediction), Align GVGD (Align Grantham Variation/Grantham Deviation), and PROVEAN (Protein Variation Effect Analyzer).

## 3. Results

### 3.1. Identification of Pathogenic Changes in OI Families

As a result of the research, we identified 29 mutations in 4 genes (*COL1A1, COL1A2, IFITM5, P3H1*) responsible for the development of OI in 42 patients from 32 families. Thus, 16 mutations are in the *COL1A1* gene, 11 in the *COL1A2* gene, 1 in the *P3H1* gene, and 1 in the *IFITM5*; 7 patients had genetic defects typical of other disorders, and 13 patients did not have any pathogenic mutations.

On the first stage of our research we identified mutations in the *COL1A1* gene (c.358C>T, c.375dupC, c.407dupG, c.579delT, c.658C>T, c.858+1G>A, c.967G>T, c.1081C>T, c.1243C>T, c.2444delG, c.2461G>A, c.2569G>T, c.2869C>T, c.3076C>T, c.3792delG, c.1354-12G>A) and the *COL1A2* gene (c.647G>A, c.874G>A, c.1197+5G>A, c.1826G>A, c.1897_1902dupGCTGGT, c.2341G>C, c.2756G>A, c.2971G>C, c.3034G>A, c.3277G>A, c.3977A>G). At the second stage, we found mutations in genes *P3H1* (c.1051G>T) and *IFITM5* (c.-14C>T). At the third stage, we found mutations in genes *FGF23* (c. 212G>C), *CLCN7* (c.141+4A>C), and *TGFB1* (c.945G>C). At the fourth stage, we identified mutations in *LAMB3* (c.1903C>T), *ALOX12B* (c.526G>A), *PLEKHM1* (c.2902-9C>T), *ERCC4* (c.2395C>T), *ARSB* (c.454C>T), and *PTH1R* (c.342C>A).

In our research, 93.1% of all identified mutations were found in I type of collagen genes (*COL1A1/COL1A2*). The mutations in the type I collagen genes, leading to OI, are usually divided into two groups: those leading to a decrease in the amount of protein (quantitative mutations or mutations of haploinsufficiency) and mutations that cause structurally abnormal protein chains (qualitative or structural mutations, mostly glycine substitutions in the triple helix collagen type I). Most of the mutations (87.5%, n = 14/16) in the *COL1A1* gene detected in the patients we examined turned out to be haploinsufficiency mutations, which lead to a mild course of the disease. On the contrary, structural mutations were mostly found in the *COL1A2* gene, accounting for 81.8% (n = 9/11) (Table 1).

Since alpha2 chains are only 1/3 of the type I collagen protein, the effect of mutations is similar to quantitative mutations in the *COL1A1* gene.

Pathogenic changes in the *COL1A1* and *COL1A2* genes mostly occurred in a single variant except for the c.579delT and c.3076C>T mutations in *COL1A1*, each of which was found in two unrelated families of Tatar ethnicity. In *COL1A1*, we identified 7 nonsense mutations, 2 splice-site mutations, 5 frameshift mutations, and 2 missense mutations, 3 of which had not been previously described in the literature. In *COL1A2*, we found 9 missense mutations, 1 frameshift mutation, and 1 splice site mutation. Seven mutations were detected for the first time and are not found in the current databases (e.g., gnomAD, Ensemble, Clinvar) (Appendix A).

Two mutations were identified in the non-collagen genes associated with the development of OI: in the *P3H1* gene, which is involved in the post-translational modification of collagen, and in the *IFITM5* gene, which is involved in the regulation of connective tissue mineralization (Table 2).

In seven patients, we identified mutations in a heterozygous state in the genes *CLCN7, ALOX12B, PLEKHM1, ERCC4, ARSB, PTH1R*, and *TGFB1*, which are associated with other connective tissue and bone diseases (Table 3). All patients had low-traumatic fractures.

Patients with mutations c.141+4A>G in the *CLCN7* gene and c.2902-9C>T in the *PLEKHM1* gene had multiple fractures as well as sensorineural hearing loss, myopia, arthropathies, and ligamentous apparatus lesions. A patient with a c.945G>C mutation in the *TGFB1* gene, associated with Kamurati-Engelman disease, had moderate osteoporosis along with muscle pain syndrome. The patient with the c.526G>A mutation in the *ALOX12B* gene, in addition to fractures, had keratinization of the skin and congenital hydronephrosis. The carrier of the c.2395C>T mutation in the *ERCC4* gene, associated with the development of Fanconi anemia, had no marked clinical manifestations, which is most likely due to the young age of the patient (at the time of examination, she was 5 years old). A patient with the c.342C>A mutation in the *PTH1R* gene had multiple manifestations of connective tissue damage, such as knee ligament rupture, congenital heart valve defects, osteoporosis, and scoliosis, which, in general, is characteristic of chondrodysplasia. A patient with a c.454C>T mutation in the *ARSB* gene, typical of type VI mucopolysaccharidosis, had osteoporosis and multiple fractures of peripheral bones. These mutations, which are not currently recognized as responsible for the development of OI, result in alterations of bone quality.

Among the 13 patients who did not have mutations, 2 patients had secondary bone lesions in endocrinological pathology (thyroid diseases and type I diabetes mellitus) and two more had signs of juvenile osteoporosis without systemic connective tissue damage. In five patients, during dynamic follow-up, the diagnosis OI was not confirmed. In four patients with a vivid clinical picture of their lesions, it is still necessary to continue the research to identify the molecular cause of the disease.

Thus, out of 62 patients from 52 families examined in this study, 9 patients had the diagnosis of OI excluded as a result of retrospective clinical observation as well as the absence of pathogenic changes in the targeted genes of bone metabolism. Considering that we detected changes in genes that are not causal genes of OI in 7 patients, they were also placed under the dynamic supervision of the clinical geneticists without the diagnosis of OI. This indicates the need for a more detailed approach to the clinical stage of diagnosis. According to the results of the study, the clinical diagnosis was confirmed in 42 patients from 32 unrelated families, and the search for pathogenic changes will continue in 4 patients. In our sample of examined patients from the Republic of Bashkortostan, 61.5% (32/52) families of patients had mutations in the genes responsible for the development of OI, 13.5% of patients had structural changes in other genes, and 25% of patients had no changes. Further calculations were performed for the 42 patients from 32 families.

### 3.2. Clinical and Genetic Characteristics of Patients with Identified Mutations

In total, in our patient population with OI, we detected 27 mutations in the *COL1A1/COL1A2* genes; 10 of them were not previously described in the literature, which is 37% (10/27) of all identified mutations in these two genes. According to our data, the share of structural changes in the *COL1A1/COL1A2* genes accounted for 40.7% of mutations (11/27); 59.3% of mutations (16/27) turned out to be haploinsufficiency variants; 8 of the 27 mutations in the *COL1A1/COL1A2* genes were caused by glycine substitutions (2 mutations in the *COL1A1* gene and 6 mutations in the *COL1A2* gene).

The most common in our sample of patients were serine substitutions (n = 4, 44.4%), followed by arginine (n = 2, 22.2%), asparagine (n = 1, 11.1%), and cysteine (n = 1, 11.1%) substitutions. Three missense mutations in the *COL1A2* gene were arginine substitutions for histidine at the 216 position and for glutamine at the 609 position of the protein, as well as lysine replacement for arginine at the 1326 position. Mutations of c.874G>A (p. Gly292Ser), c.647G>A (p. Arg216His), and c.2341G>C (p. Gly781Arg) of the *COL1A2* gene led to a mild course of the disease with type I of OI; c.2461G>A (p. Gly821Ser), c.2569G>T (p. Gly857Cys) mutations of the *COL1A1* gene and c.2756G>A (p. Gly919Asp), c.3277G>A (p. Gly1093Ser) mutations of the *COL1A2* gene to type III; and mutation c.2971G>C (p. Gly991Arg) of the *COL1A2* gene to type IV of OI.

In our sample, family cases are 31.3% (10 families, 10/32). In the Chinese population, family cases are 33% [12], in Italians 32% [13] and 53% in Taiwanese [14]. According to our results, in families with mutations c.579delT, c.967G>T, c.1354-12G>A of the *COL1A1* gene, the phenotypes of the probands and parents coincided and led to type I of OI. The c.3034G>A mutation of the *COL1A2* gene was detected in the proband and her mother with type III of OI. Mutation c.3977A>G resulted in type IV of OI, both in the proband and in one of the parents. However, in patients with c.1081C>T, c.2869C>T mutations in the *COL1A1* gene and in patients with mutations c.1826G>A, c.2756G>A, c.2971G>C of the *COL1A2* gene, clinical signs of the disease differed within the same family. In the proband, the same mutation led to severe phenotypic signs of the disease, unlike parents who do not have signs of OI, which most likely indicates the incomplete penetrance of the parents. All identified mutations in type I collagen genes were unique for each family, except for two mutations, c.579delT and c.3076C>T in the *COL1A1* gene, each of which was found in two unrelated families with Tatar ethnicity.

In two patients with type I of OI, we identified complex heterozygotes by mutations in the *COL1A1* and *COL1A2* genes. A patient of Russian ethnicity had a mutation of c.2869C>T in the *COL1A1* gene and a splicing mutation of c.1197+5G>A in the *COL1A2* gene. The same c.1197+5G>A mutation of the *COL1A2* gene was detected in a heterozygous state with a c.579delT mutation of the *COL1A1* gene in a patient of Tatar ethnicity. Mutations c.579delT, c.2869C>T of the *COL1A1* gene and c.1197+5G>A of the *COL1A2* gene were previously described in patients with OI and are pathogenic [4,5,12,14,15,16,17,18,19]. A patient of Russian origin with type IV of OI was found to have a combined mutation of c.2971G>C in the *COL1A1* gene and c. 212G>C (p. Ser71Thr) in the *FGF23* gene. Patient 40 had a combined mutation of c.-14C>T in the *IFITM5* gene and c.1903C>T, p. Arg635* in the *LAMB3* gene (Table 4).

The c.579delT mutation (p. Gly194fs) was found in two unrelated probands born in 2002 and 1995. Both patients had blue sclera, multiple deformities, and hypermobility of joints. One patient additionally had brachycephaly and imperfect dentinogenesis. This sequence change creates a signal of a premature stop of translation (p. Gly194fs) in the *COL1A1* gene. This mutation c.579delT was described 25 times in the database of OI and identified in populations of Spaniards, Italians, Swedes, Ukrainians, Chinese, and Taiwanese [4,12,13,14,16,17,19,20]. This mutation leads to type I of OI, with a relatively mild course of the disease.

Mutation c.3076C>T was also detected in two unrelated patients. Both patients were males with type I of OI, which is characterized by blue sclera and mild deformities of the bone system. This mutation has been described 14 times in patients from the USA, Great Britain, Italy, Brazil, China, and Finland with type I of OI [12,14,18,21,22,23,24,25,26]. Mutations c.358C>T, c.579delT, c.658C>T, c.967G>T, c.1081C>T, c.1243C>T, c.2869C>T, c.2444delG, c.3076C>T, c.858+1G>A, c.1354-12G>A, c.2461G>A, c.2569G>T of the *COL1A1* gene and mutations c.874G>A, p. 1197+5G>A, c.2756G>A, c.3034G>A of the *COL1A2* gene were published earlier in the database [6].

In the *P3H1* gene, which is involved in the post-translational modification of collagen, we found a c.1051G>T (p. Glu351*) mutation in a male patient of Bashkir ethnicity. This mutation was detected for the first time. The proband was born in a consanguineous marriage and was characterized by white sclera, a round face, deformities of the lower extremities, and disability to walk independently (he has been using a wheelchair since his early childhood). The parents of the proband and sibs are healthy.

In the *IFITM5* gene, we identified the c.-14C>T mutation in three unrelated patients at once. In two patients, we observed all clinical signs of OI typical of this type: calcification of the interosseous membrane, displacement of the radial head, deformities of the shins, femurs, and knee joints. In one Russian patient, we identified two mutations simultaneously—one in the *IFITM5* gene, which is a causal mutation of type V, and the other—c.1903C>T (p. Arg635*)—in the *LAMB3* gene, responsible for the manifestation of epidermolysis bullosa. This patient was found to have 15 fractures, blue sclera, barrel chest, triangular face, kyphoscoliotic deformity of the spine, deformities of the lower legs, and imperfecta dentinogenesis (Table 4).

## 4. Discussion

### 4.1. Mutations of the COL1A1 Gene

To date, we know more than 1000 structural changes in the *COL1A1* gene. Structural mutations in this gene account for about 45%, and the remaining number of mutations are accounted for by other variants (nonsense mutations, reading frame shift mutations, splicing site mutations, deletions of the entire gene). According to literature data, the percent of new pathogenic mutations in two type I collagen genes (*COL1A1/COL1A2*) in Ukrainians with OI was 42.85%, in Chinese—40.98%, and in Swedes—31.53% [4,12,19]. In the Chinese population, structural changes account for 54%, and haploinsufficiency mutations account for 46% [12]; in the Ukrainian population, the ratio is exactly 49%/51% [19], which differs from our results. We had structural mutations of 41%, and haploinsufficiency mutations accounted for 59%.

Studies of the three-spiral peptide have shown that a simple replacement of glycine in the medium (Gly-Pro-Hyp) strongly destabilizes the chain and affects the clinical severity of osteogenesis imperfecta. We also determined that the degree of destabilization of the peptide depends on that which amino acid replaces glycine (Gly). The order of stability loss, from the smallest to the largest, is as follows: Ala ≤ Ser < Cys < Arg < val < Glu ≤ Asp [27,28]. Substitutions of glycine with alanine and serine have the least effect on the conformation and stability of the peptide and lead to a lighter course of the disease [28]. Furthermore, the degree of destabilization depends on the location of the mutation site [29].

In lethal types of OI, glycine substitutions for Val, Asp, Glu and Arg are observed more often. On the contrary, glycine substitutions for serine and cysteine are rare in patients with fatal cases of OI [5,30,31].

According to the literature data, Gly residues are most often replaced by Ser or Cys [28].

Mutations c.358C>T, c.658C>T, c.1243C>T, c.2869C>T, c. 3076C>T, c.858+1G>A, c.1354-12G>A, and c.3208-1G>C were detected in patients with type I of OI. Patients from other populations had similar clinical manifestations to patients from our study. Mutations c.1081C>T, c.2461G>A, and c.2569G>T found in the *COL1A1* gene lead to severe clinical symptoms in our patients with type III of OI than in patients from literature who had type I of OI [6].

Thus, the replacement of cytosine with thymine in the 1081 position of cDNA, leading to a stop codon in the proband from the Republic of Bashkortostan, led to severe clinical manifestations of the disease with type III of OI. He had multiple fractures, which led to deformities of the limbs and disability. The father with the same mutation had type I of OI. This change has been described 11 times, and the authors report a mild course of the disease with type I of OI [4,12,32,33,34,35]. The c.2461G>A mutation was registered in the database 31 times and described in patients with types I, II, III, and IV of OI [4,5,12,13,14,17,36,37,38,39,40,41]. Kloen and co-authors described in detail a patient with this mutation who had multiple fractures with poor healing as a result of this progressive deformation of the lower and upper extremities and compression fractures of the spine. The patient could not move independently [40]. Our patient also has multiple fractures and deformities of the limbs, which correlates with type III of OI.

The c.2569G>T mutation in the *COL1A1* gene was described 8 times [5,42]. The phenotypes of patients also differed (II, III, IV types of OI). In our patient, this change led to type III of OI. The patient had multiple fractures, short stature, and deformities of the bone system.

### 4.2. Mutations of the COL1A2 Gene

There are about 600 mutations described in the *COL1A2* gene. According to the international database on osteogenesis imperfecta, the vast majority of mutations in the *COL1A2* gene are missense mutations, which account for approximately 74% [6].

It is noted that the most frequent structural defects of type I collagen causing osteogenesis imperfecta are glycine substitutions in the spiral domain. Glycine substitutions delay helical folding, increasing the access time for enzyme modification. Thus, in our sample of OI patients, the most frequent missense mutations were glycine substitutions, which accounted for 73% of all structural changes identified in collagen genes.

Mutations in different type I collagen chains differ in their phenotypic effect. In the α1 chain of collagen type I, substitutions with charged or branched side chains disrupt the stability of the spiral and are predominantly lethal. Substitutions in the two main ligand-binding regions near the carboxyl end of the α1 chain have exceptionally lethal outcomes, indicating important interactions between the collagen monomer and non-collagen matrix proteins. In the α2 chain of type I collagen, substitutions are mostly non-lethal; however, eight lethal clusters along the chain align with proteoglycan binding sites on collagen fibrils. Finally, less than 5% of the mutations causing classical osteogenesis imperfecta occur in the procollagen C-propeptide, disrupting chain association or folding [43].

The c.874G>A mutation detected in the *COL1A2* gene was published 6 times [4,20,41] and resulted in type I in 5 patients from Sweden and Germany [4,20] as well as a patient from Belarus. However, Duy described that this change led to type III of OI in a patient from Vietnam [41].

The c.2756G>A mutation of the *COL1A2* gene was previously published in a patient with intrauterine fractures and various anomalies with type II of OI [44]. Our patient had short stature, blue sclera, and multiple fractures, which led to disability of the patient and progressive deformities of the lower extremities with type III of OI.

The c.3034G>A mutation has been published 26 times in the database on OI [4,5,38,39,41,43,45,46,47]. Patients with this change had III and IV types of OI. According to clinical signs, our patient was classified as type III of OI with blue sclera and multiple deforming fractures.

According to literature data, haploinsufficiency mutations in *COL1A1/COL1A2* genes resulting from splicing site mutations, meaningless mutations, deletions, or insertions usually create a premature termination codon. These aberrant RNAs are usually decomposed by nonsense-mediated mRNA decay (NMD). With normal type I collagen α chains produced by the wild-type allele, haploinsufficiency usually leads to a moderate OI phenotype [5]. On the other hand, the dominant negative effects are the result of missense mutations or mutations of the premature terminating codon, which avoid nonsense-mediated mRNA decay. Binding of the mutated α chain with normal α chains produced by the wild-type allele leads to an abnormal type of collagen I. Classically known OI mutations with a dominant negative effect represent the substitution of an amino acid for one of the mandatory glycine residues. They are found in every third position in the COL1A1 chain and other mutations, such as defects in the passage of exons or deletions in the reading frame. Most cases of OI with dominant negative effects are usually more serious than cases of haploinsufficiency [48].

We hope that the mutations in patients with OI and the clinical characteristics that we have described will contribute to the understanding of genotypic–phenotypic correlations.

### 4.3. Mutations in the P3H1 Gene

The phenotype of our patient is similar to the patients described by Baldridge et al. (2008), who noted that mutations in this gene lead to different clinical characteristics compared to patients with mutations in type I collagen genes [49]. Patients are characterized by white sclera, round face, and deformities of the lower extremities. As the researchers note, zero mutations in the *P3H1* gene cause type III of OI and are severe or lethal and lead to excessive modification of the entire spiral region of collagen. The P3H1 enzyme itself causes 3-hydroxylation of α1(I) Pro986 in type I collagen as part of a complex of three proteins (P3H1, CRTAP, and CypB) in a ratio of 1:1:1. P3H1 is a catalytically active component, whereas CRTAP is an auxiliary protein without a catalytic domain. Prolyl-3-hydroxylation is one of several modifications of pro-chains that contribute to the proper stacking, stability, and secretion of procollagen. Prolyl-4-hydroxylation is important for the thermal stability of the triple helix, while lysine hydroxylation and hydroxylysine glycosylation contribute to the extracellular stability of cross-links between molecules [50]. Zero mutations of P3H1 actually cancel the 3-hydroxylation of type I collagen. The absence of hydroxylation of α1(I) Pro986 and/or the direct chaperone effect of P3H1 leads to a delay in the folding of the collagen spiral [51]. Patients with mutations in this gene have been described in African Americans, Africans, Pakistanis, and Arabs. The researchers note that mutations in this gene most often occurred in consanguineous patients. Pepin et al. 2013 described founder-effect mutations in the African population [49,51,52]. Zero mutations in *P3H1* or *CRTAP* lead to the absence of both proteins in mutant cells because these proteins are mutually supportive in the complex and ultimately lead to similar clinical signs in patients with OI.

### 4.4. Mutation of the IFITM5 Gene

Type V of OI is unique among all types of osteogenesis imperfecta: most patients (approximately 95%) with type V have the same heterozygous mutation in IFITM5, a point mutation in 5′-UTR (c.-14C>T), which generates a new starting codon and adds five residues to the N–terminal end of the protein. Patients with this type have moderate bone dysplasia with a different combination of distinctive features, including ossification of the interosseous membrane of the forearm (76–100%) and dislocation of the radial head (36–88%) and its displacement (86%) [53]. More than half of patients with type V develop hyperplastic corns during fracture healing. The sclera hue varies, and the teeth remain normal. All patients with type V have a distinct mesh plate on bone histology [54,55,56,57,58].

## 5. Conclusions

Thus, pathogenic mutations responsible for the development of OI were found in 32 unrelated families. We identified 16 pathogenic changes in the *COL1A1* gene, 11 pathogenic mutations in the *COL1A2* gene, and 1 mutation each in the *P3H1* and *IFITM5* genes. *COL1A1* is 55.2% (16/29) of the detected mutations, 37.9% (11/29) from the *COL1A2* gene, and 3.45% (1/29) each for the *IFITM5* and *P3H1* genes. Structural changes in gene characteristics of other diseases were detected in 13.46% (7/52) of patients and no changes in 25% (13/52) of the patients. In our research, 93.1% of all identified mutations were found in type I of the collagen genes (*COL1A1/COL1A2*).

Among patients with identified mutations involved in the development of OI, we revealed autosomal dominant inheritance in 10 families; mutations occurred de novo in 15 families and in 7 families—the parents’ DNA was not available.

We found 11 previously undescribed pathogenic changes: 3 in the *COL1A1* gene, 7 in *COL1A2*, and 1 mutation in the *P3H1* gene. The identified mutations turned out to be unique, with the exception of mutations c.579delT and c.3076C>T in the *COL1A1* gene found in two unrelated families as well as mutations c.-14C>T in the *IFITM5* gene found in three unrelated patients.

## Figures and Tables

**Figure 1 genes-13-00124-f001:**
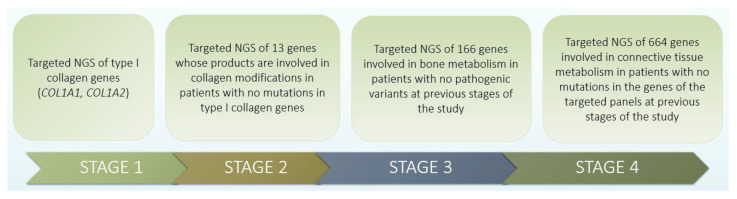
Four stages of research design.

**Table 1 genes-13-00124-t001:** Clinical characteristics of patients with mutations in type I collagen genes.

Family	Patient	Mutation	Inheritance Type	OI Type	Sex	Age	Blue Sclera	Fractures
** *COL1A1* **								
**Family 1**	Patient 1	c.358C>T, p. Arg120*	de novo	1	F	68	+	18
**Family 2**	Patient 2	c.375dupC, p. Ala126fs	de novo	1	F	54	+	11
**Family 3**	Patient 3	c.407dupG, p.Gly136fs	de novo	1	F	5	+	3
**Family 4**	Patient 4	c.579delT, p.Gly194fs	de novo	1	M	19	+	33
**Family 5**	Patient 5	c.579delT, p.Gly194fs	AD *	1	M	26	+	15
	Patient 6			1	F	59	−	3
**Family 6**	Patient 7	c.658C>T, p.Arg220*	no parents’ DNA	1	M	9	+	11
**Family 7**	Patient 8	c.858+1G>A	no parents’ DNA	1	M	32	+	15
**Family 8**	Patient 9	c.967G>T, p.Gly323*	AD	1	F	30	+	8
	Patient 10			1	F	52	+	20
**Family 9**	Patient 11	c.1081C>T, p.Arg361*	AD	3	M	29	+	19
	Patient 12			3	M	52	−	−
**Family 10**	Patient 13	c.1243C>T, p.Arg415*	de novo	1	F	28	+	10
**Family 11**	Patient 14	c.2444delG, p.Gly815fs	no parents’ DNA	1	F	27	+	13
**Family 12**	Patient 15	c.2461G>A, p.Gly821Ser	no parents’ DNA	3	F	28	+	15
**Family 13**	Patient 16	c.2569G>T, p.Gly857Cys	no parents’ DNA	3	F	12	+	50
**Family 14**	Patient 17	c.2869C>T, p.Gln957*	AD	1	F	17	+	14
	Patient 18			1	M	46	−	−
**Family 15**	Patient 19	c.3076C>T, p.Arg1026*	de novo	1	M	9	+	7
**Family 16**	Patient 20	c.3076C>T, p.Arg1026*	no parents’ DNA	1	M	17	+	12
**Family 17**	Patient 21	c.3792delG, p.Met1264fs	de novo	4	M	6	−	4
**Family 18**	Patient 22	c.1354-12G>A	AD	1	M	10	+	6
	Patient 23			1	F	36	−	3
** *COL1A2* **								
**Family 1**	Patient 24	c.647G>A, p. Arg216His	no parents’ DNA	4	M	12	−	3
**Family 2**	Patient 25	c.874G>A, p. Gly292Ser	no parents’ DNA	1	M	20	+	7
**Family 3**	Patient 26	c.1826G>A, p. Arg609Gln	AD	1	M	11	+	5
	Patient 27			1	M	41	−	−
**Family 4**	Patient 28	c.1897_1902dupGCTGGT, p. Ala633_Gly634dup	no parents’ DNA	1	F	20	+	5
**Family 5**	Patient 29	c.2341G>C, p. Gly781Arg	de novo	1	M	4	+	5
**Family 6**	Patient 30	c.2756G>A, p. Gly919Asp	AD	3	M	23	+	10
	Patient 31			3	F	52	−	−
**Family 7**	Patient32	c.2971G>C, p. Gly991Arg	AD	4	M	34	−	7
	Patient 33			4	M	58	−	5
**Family 8**	Patient 34	c.3034G>A, p. Gly1012Ser	AD	3	F	6	+	5
	Patient 35			3	F	32	+	7
**Family 9**	Patient 36	c.3277G>A, p. Gly1093Ser	de novo	1	F	14	+	90
**Family 10**	Patient 37	c.3977A>G, p. Lys1326Arg	AD	4	M	4	−	4
	Patient 38			4	M	45	−	−

* AD—autosomal-dominant

**Table 2 genes-13-00124-t002:** Clinical characteristics of patients with mutations in the *P3H1* and *IFITM5* genes.

Family	Patient	Mutation	Inheritance Type	OI Type	Sex	Age	Blue Sclera	Fractures
***P3H1* mutations**								
**Family 1**	Patient 39	c.1051G>T, p. Glu351*	de novo	3	M	24	+	12
***IFITM5* mutations**								
**Family 2**	Patient 40	c.-14C>T	de novo	5	F	27	+	15
**Family 3**	Patient 41	c.-14C>T	de novo	5	F	26	+	50
**Family 4**	Patient 42	c.-14C>T	de novo	5	M	10	+	10

**Table 3 genes-13-00124-t003:** Clinical characteristics of patients with mutations in non-collagen genes.

No.	Patient	Mutation	Inheritance Type	Age	Sex	Blue Sclera	Number of Fractures	Disorders
**1**	Patient 43	*ALOX12B*: c.526G>A, p. Glu176Lys	de novo	12	M	-	4	Hyperkeratosis, metatarsal and metacarpal fractures
**2**	Patient 44	*PLEKHM1*: c.2902-9C>T	de novo	15	F	+	>9	Sensorineural hearing loss, decreased Bone mineral density (BMD), myopia, arthropathy
**3**	Patient 45	*ERCC4*: c.2395C>T, p. Arg799Trp	de novo	5	F	+	1	Simple farsighted astigmatism, mild anemia
**4**	Patient 46	*ARSB*: c.454C>T,p. Arg152Trp	de novo	12	M	+	18	He started talking late. The gait is limping. There are no visible deformations.
**5**	Patient 47	*PTH1R*: c.342C>A, p. His114Gln	de novo	8	M	+	>5	BMD decrease, kyphoscoliotic deformity of the chest, fractures of the metacarpal bones
**6**	Patient 48	*CLCN7*: c.141+4A>C	de novo	31	M	+	>11	Decrease in BMD, facial phenotypes (frontal bumps), fractures of the bones of the hand, foot
**7**	Patient 49	*TGFB1*: c.945G>C, p. Lys315Asn	de novo	29	F	+	>17	Short stature, varus deformity of the limbs, sabre-shaped deformity of the shins, kyphoscoliotic deformity of the chest, curvature of the left forearm, high palate.

**Table 4 genes-13-00124-t004:** Clinical characteristics of patients with combined mutations.

	Family /Patient	Mutation 1	Mutation 2	Inheritance Type	Age	OI Type	Sex	Blue Sclera	Fractures	Disorders
**1**	Patient 17	c.2869C>T, p. Gln957* in *COL1A1* gene	c.1197+5G>A in *COL1A2* gene	AD *	17	1	F	+	14	Knee joint deformities, saber-shaped deformity of the hips
**2**	Patient 4	c.579delT, p. Gly194fs in *COL1A1* gene	c.1197+5G>A in *COL1A2* gene	de novo	19	1	M	+	33	Hypermobility of joints, deformity of the elbow joint
**3**	Patient 33	c.2971G>C, p. Gly991Arg in *COL1A2* gene	c.212G>C, p.Ser71Thr in *FGF23* gene	AD	58	4	M	−	5	Short stature
**4**	Patient 40	c.-14C>T in *IFITM5* gene	c.1903C>T, p. Arg635* in *LAMB3* gene	de novo	27	5	F	−	15	Kyphoscoliotic deformity of the spine and chest, platyspondylia, hypermobility of joints, arched curvature of the bones of both legs

* AD-autosomal-dominant

## Data Availability

Not applicable.

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
