# Peer review of "Osteogenesis Imperfecta: Search for Mutations in Patients from the Republic of Bashkortostan (Russia)"

_genes, 2022, doi:10.3390/genes13010124_

Round 1

Reviewer 1 Report

The study reported next-generation sequencing results from 62 patients with Osteogenesis Imperfecta from Russia. The manuscript increases the knowledge in the field of rare bone diseases, as data from Russia on this disease are rare and of high interest for the medical scientific community.

Introduction

- Lines 28-29: please include the reference

- Lines 40-45: please include reference

- Lines 50-51: “The aim of the work was to search for pathogenic changes in target genes in patients from the Republic of Bashkortostan with Osteogenesis Imperfecta by NGS-technology”

- Did all patients have a clinical diagnosis of OI? If yes, please include the information.

Materials and Methods

- Line 53: please substitute “middle” for “mean’ age

- Lines 53-54: 62 patients were selected as candidates for NGS of how many?

- Lines 55-59: What do you mean by “clinical manifestations characteristic of osteogenesis incomplete”? The diagnosis of OI is based on clinical features and radiological findings. Please be clearer about the selection of the sample for the NGS study and previous diagnosis information on these individuals (genetic tests/clinical info).

- How “number of fractures” was documented? Was it patient-reported or radiographically confirmed?

- Lines 66-67 + figure 1: How did the genes for the panels for NGS were selected? How the protocol to sequencing 166 or 664 genes were created? Which references, databases or platforms were used?

- Lines 91-92: “at the second stage, targeted NGS sequencing of 13 genes was carried out”. Please check the information as the figure 1 reports 18 genes in stage 2.

- As every stage of the analysis was carried out in different laboratories, how did the authors guarantee the storage, distribution and quality of the biological material collected?

- Line 145: please correct the phrase to “OI database” as only one database was used as reference.

Results

- Please harmonize the presentation of the data reported, it is confusing as it is now. I suggest to report the results as described in the methodology, divided into 4 stages.

- As explicit in the title all individuals included in the study are from the Republic of Bashkortostan, please avoid to repeat this information constantly as it is already implicit.

Conclusion

- I suggest to highlight in the conclusion that most of the mutations found by the authors are consistent with previously known OI main target genes (COL1A1 gene COL1A2 gene).

Table 4

Patient 33. Did the patient had only short stature and 5 fractures during whole life as clinical manifestation? Why did he receive an OI type 4 diagnosis? Only short stature is not indication of OI.

Patient 40. Did this patient have any feature of epidermolysis bullosa?

Author Response

Dear reviewer,

Thank You for attention to our manuscript and valuable recommendations. We accepted all your remarks and corrected text according to your recommendations.

Point 1: - Lines 28-29: please include the reference

Response 1: We included reference:

The total frequency of the disease in general varies from 1:15,000 to 1:20,000 [Forlino A, Marini JC. Osteogenesis imperfecta. Lancet. 2016;387(10028):1657-1671. doi:10.1016/S0140-6736(15)00728-X]

Point 2: - Lines 40-45: please include reference

Response 2:

We included reference:

Less than 10% patients with OI have recessive forms of inheritance caused by mutations in genes encoding proteins that are involved in the synthesis, transport, and post-translational  modifications of collagen, or factors associated with differentiation and mineralization of  bone cells (CRTAP, PPIB, BMP1, CCDC134, CREB3L1, FAM46A, FKBP10, P3H1, P4HB,  PLOD2, PLS3, SEC24D, SERPINF1, SERPINH1, SP7, SPARC, TMEM38B)

[1.  Forlino A, Marini JC. Osteogenesis imperfecta. Lancet. 2016;387(10028):1657-1671. doi:10.1016/S0140-6736(15)00728-X;

  1. Zaripova A.R., Khusainova R.I. Modern classification and molecular-genetic aspects of Osteogenesis Imperfecta. Vavilov Journal of Genetics and Breeding. 2020. 24 (2), 219-227; DOI 10.18699/VJ20.614
  2. https://oi.gene.le.ac.uk/home.php].

Point 3

- Lines 50-51: “The aim of the work was to search for pathogenic changes in target genes in patients from the Republic of Bashkortostan with Osteogenesis Imperfecta by NGS-technology”

- Did all patients have a clinical diagnosis of OI? If yes, please include the information.

Response 3:

All patients had a diagnosis of Osteogenesis imperfecta, presented on the basis of clinical criteria and radiological data. According to the long-term clinical observation and molecular research, the diagnosis was excluded in some patients, the results are presented in the text 

Point 4

Materials and Methods

- Line 53: please substitute “middle” for “mean’ age

Response 4: We changed “middle” for “mean’ age

Point 5

- Lines 53-54: 62 patients were selected as candidates for NGS of how many?

Response 5:

62 patients was the total sample at the beginning of 2020. We corrected description of study material in the text.

Point 6

- Lines 55-59: What do you mean by “clinical manifestations characteristic of osteogenesis incomplete”? The diagnosis of OI is based on clinical features and radiological findings. Please be clearer about the selection of the sample for the NGS study and previous diagnosis information on these individuals (genetic tests/clinical info).

Response 6:

The time of occurrence of the first low-traumatic fracture was taken as the clinical manifestation of osteogenesis imperfecta, after which the patient was clinically examined, including with the use of X-ray diagnostics.

- How “number of fractures” was documented? Was it patient-reported or radiographically confirmed?

It was patient-reported with radiological confirmation

Point 7

- Lines 66-67 + figure 1: How did the genes for the panels for NGS were selected?

Response 7:

On the first stage NGS panel included two genes. On the second stage – 13 genes, the third stage – 166 genes, on the fourth stage – 664 genes were observed.

NGS PANEL OF THE FIRST STAGE

COL1A1, COL1A2

NGS PANEL OF THE SECOND STAGE

BMP1, COL1A1, COL1A2, CRTAP, FKBP10, IFITM5, LEPRE1, PPIB, SERPINF1, SERPINH1, SP7, TMEM38B, WNT1

NGS PANEL OF THE THIRD STAGE

ADAMTSL2, AGPS, ARSB, ARSE, ATP6V1B1 (ATP6B1), ATP6V0A4, BMP1, CA2, COL10A1, COL9A3, CREB3L1, CRTAP, CTSA, CYP27B1, CYP2R1, DDR2, DMP1, DYNC2H1, EHHADH, FAH, FUCA1, GALNS, GNPAT, GNPTG, GNS, GPX4, GUSB, IDS, IFITM5, IFT122, IFT140, IFT43, IFT80, INPPL1, LEPRE1, LIFR, LONP1, MAN2B1, MANBA, MCOLN1, MMP9, NEK1, NKX3-2, PAPSS2, PHEX, PLOD2, PLS3, PPIB, RAB33B, SBDS, SERPINF1, SGSH, SLC34A3, SLC35D1, SLC4A4, SMARCAL1, SP7, SPARC, SUMF1, TMEM38B, TRIP11, WDR34, WDR60, TRIM37, TCIRG1, ROBO3, DLL3, TRAPPC2, ACVR1, HEXB, HYAL1, GM2A, AP2S1, SLC9A3R1, MTAP, ALPL, ANKH, CLCN5, COL11A1, COL11A2, COL1A1, COL1A2, COL2A1, COL9A1, COL9A2, COMP, DYM, EBP, ENPP1, EVC, EVC2, FBN1, FGF23, FGFR3, FKBP10, FLNA, FLNB, GLB1, GNPTAB, HGSNAT, HNF4A, HSPG2, IDUA, IFT172, IL1RN, LBR, LRP5, MATN3, MMP13, NAGLU, NEU1, NME1, NSDHL, PEX1, PEX5, PEX6, PEX7, PHOSPHO1, PTH1R, RUNX2, SERPINH1, SLC17A5, SLC26A2, SLC2A2, SLC34A1, SLC4A1, SOX9, SPP1, TCTN3, TRPV4, TTC21B, VDR, WDR19, WDR35, WNT1, RMRP, FBN2, MYH3, ROR2, BTK, GDF6, GNAS, MSX2, GLI3, FGFR1, FGFR2, HOXD13, ERCC6, HEXA, CASR, GNA11, CLCN7, GJA1, HPGD, OSTM1, PLEKHM1, PTDSS1, SLCO2A1, SNX10, SOST, TBXAS1, TGFB1, TNFRSF11A, TNFRSF11B, TNFSF11, TYROBP

NGS PANEL OF THE FOURTH STAGE

AAGAB, ABCA12, ABCB6, ABCC6, ABCC9, ABHD5, ACP5, ACTA2, ACVR1, ACVR2B, ACVRL1, ADAM10, ADAMTS10, ADAMTS2, ADAMTSL2, ADAR, AEBP1, AGPS, AKT1, AKT3, ALDH18A1, ALDH3A2, ALMS1, ALOX12B, ALOXE3, ALPL, ALX4, AMER1, ANKH, ANO5, AP1S1, AP2S1, AP3B1, APCDD1, AQP5, ARSB, ARSE, ATM, ATP2A2, ATP2C1, ATP6V0A2, ATP6V1A, ATP6V1E1, ATP7A, ATR, AXIN2, B3GALT6, B3GAT3, B4GALT7, BANF1, BCS1L, BGN, BHLHA9, BLM, BLOC1S3, BLOC1S6, BMP1, BMP2, BMPER, BMPR1B, BRAF, C1R, C1S, CA2, CANT1, CARD14, CASR, CBL, CBS, CCBE1, CCDC8, CCM2, CCND2, CDC6, CDH3, CDKN1C, CDSN, CDT1, CENPJ, CEP152, CEP63, CERS3, CFC1, CHST14, CHST3, CLCN5, CLCN7, CLDN1, COL10A1, COL11A1, COL11A2, COL12A1, COL17A1, COL1A1, COL1A2, COL2A1, COL3A1, COL4A1, COL5A1, COL5A2, COL7A1, COL9A1, COL9A2, COL9A3, COMP, CREBBP, CRELD1, CRTAP, CSTA, CTC1, CTSC, CTSK, CUL7, CYLD, CYP27B1, CYP2R1, CYP4F22, DCHS1, DDB2, DDR2, DHCR24, DHCR7, DKC1, DLL3, DLX3, DMP1, DNA2, DOK7, DSC3, DSE, DSG1, DSG4, DSP, DST, DTNBP1, DYM, DYNC2H1, EBP, ECEL1, EDN3, EDNRB, EFEMP2, EFNB1, EIF2AK3, ELN, ELOVL4, ENG, ENPP1, EP300, EPG5, ERCC2, ERCC3, ERCC4, ERCC5, ERCC6, ERCC8, ERF, ESCO2, EVC, EVC2, EXPH5, FAM111A, FAM20C, FAT4, FBLN1, FBLN5, FBN1, FBN2, FERMT1, FGF10, FGF16, FGF23, FGF9, FGFR1, FGFR2, FGFR3, FKBP10, FKBP14, FLCN, FLG, FLNA, FLNB, FLT4, FOXC2, FOXE3, FOXN1, FREM1, GALNS, GATA2, GDF1, GDF2, GDF3, GDF5, GDF6, GJA1, GJB2, GJB3, GJB4, GJB6, GJC2, GLB1, GLI3, GNA11, GNPAT, GNPTAB, GNPTG, GNS, GORAB, GPC6, GPR143, GSC, GTF2H5, GUSB, HAMP, HFE, HGSNAT, HOXA11, HOXD13, HPGD, HPS1, HPS3, HPS4, HPS5, HPS6, HR, HRAS, HSPG2, IDS, IDUA, IFITM5, IFT122, IFT140, IFT172, IFT43, IFT80, IHH, IKBKG, IL11RA, IMPAD1, INPPL1, ITGA3, ITGA6, ITGB4, JUP, KIF11, KIF22, KIT, KITLG, KRAS, KRIT1, KRT1, KRT10, KRT14, KRT16, KRT17, KRT2, KRT5, KRT6A, KRT6B, KRT6C, KRT74, KRT81, KRT83, KRT86, KRT9, LAMA3, LAMB3, LAMC2, LARP7, LBR, LEFTY2, LEMD3, LIFR, LIPH, LIPN, LMBR1, LMNA, LMX1B, LOR, LOX, LPAR6, LRP4, LRP5, LTBP3, LTBP4, LYST, MAP2K1, MAP2K2, MATN3, MBTPS2, MC1R, MED12, MEGF8, MEOX1, MFAP5, MGP, MITF, MLH1, MLPH, MMP1, MMP13, MMP9, MPLKIP, MSH2, MSH6, MSX2, MTAP, MYBPC1, MYH11, MYH3, MYH8, MYLK, MYO5A, NAGLU, NEK1, NF1, NF2, NHP2, NIN, NIPAL4, NKX2-5, NKX3-2, NODAL, NOG, NOP10, NOTCH1, NOTCH2, NPR2, NRAS, NSDHL, OBSL1, OCA2, OFD1, ORC1, ORC4, ORC6, OSTM1, P3H1, PAPSS2, PAX3, PCNT, PCYT1A, PDCD10, PDE4D, PDGFRB, PEX7, PHEX, PHYH, PIEZO2, PIK3CA, PIK3R2, PKD1, PKD2, PKP1, PLEC, PLEKHM1, PLOD1, PLOD2, PLOD3, PLS3, PMS2, PNPLA1, POC1A, POFUT1, POGLUT1, POLD1, POLH, POMP, POR, PORCN, POT1, PPIB, PRDM5, PRKAR1A, PRKG1, PTCH1, PTDSS1, PTEN, PTH1R, PTHLH, PTPN11, PYCR1, RAB23, RAB27A, RAB33B, RAF1, RAPSN, RASA1, RBBP8, RBM28, RECQL4, RHBDF2, RIN2, RIT1, RMRP, RNU4ATAC, ROR2, RTEL1, RUNX2, SALL1, SALL4, SBDS, SDHB, SDHD, SERPINB7, SERPINF1, SERPING1, SERPINH1, SF3B4, SGSH, SH3PXD2B, SHOC2, SHOX, SKI, SLC26A2, SLC27A4, SLC2A10, SLC34A1, SLC34A3, SLC35D1, SLC39A13, SLC40A1, SLC45A2, SLC9A3R1, SLCO2A1, SLURP1, SMAD3, SMAD4, SMAD6, SMARCAL1, SNAI2, SNAP29, SNRPE, SNX10, SOS1, SOST, SOX10, SOX18, SOX9, SP7, SPINK5, SPRED1, ST14, STAMBP, STK11, STS, SUMF1, TAB2, TAT, TBCE, TBX15, TBX3, TBX5, TBXAS1, TCF12, TCIRG1, TCTN3, TERC, TERT, TFR2, TGFB1, TGFB2, TGFB3, TGFBR1, TGFBR2, TGM1, TGM5, TINF2, TMEM38B, TNFRSF11A, TNFRSF11B, TNFSF11, TNNI2, TNNT3, TNXB, TP63, TPM2, TRAPPC2, TREX1, TRIP11, TRPS1, TRPV3, TRPV4, TSC1, TSC2, TTC21B, TWIST1, TYR, TYROBP, TYRP1, USB1, VDR, VEGFC, VPS33B, WDR19, WDR34, WDR35, WDR60, WISP3, WNT1, WNT10A, WNT10B, WNT5A, WNT7A, WRAP53, WRN, XPA, XPC, XYLT1, ZIC3, ZMPSTE24, ZNF469, ZSWIM6.

How the protocol to sequencing 166 or 664 genes were created? Which references, databases or platforms were used?

The first stage

A primer panel was developed for sequencing the complete gene sequence using targeted mass parallel sequencing (MPS). Enrichment of the genome regions of interest was carried out using PCR of long fragments using the BioMaster LR HS-PCR-Color (2x) kit (Biolabmix, Russia) with the production of fragments on a SureCycler 8800 amplifier (Agilent Technologies, USA). The presence of amplified products was checked by agarose gel electrophoresis. DNA libraries were prepared using the Nextera Flex kit (Illumina, USA) according to the manufacturer's recommended protocol. MPS was performed on a MiSeq sequencer (Illumina, USA) using a MiSeq Reagent Kit V2 (Illumina, USA) sequencing kit according to the protocol recommended by the manufacturer. The data obtained as a result of the MPS underwent bioinformatic processing, which included the following key stages: assessment of the quality of readings using the FastQC program, removing adapters and trimming the sequence using the Trimmomatic-0.36 package, aligning to target sequences using Bowtie2, converting to bam files, sorting them and converting them to VCF format using the Samtools set of utilities.

The study was carried out using the equipment and protocols of the "Medical Genomics" center (Tomsk, Russia) [5].

The second stage

Molecular genetic diagnostics of patients was carried out in the laboratory of the Department of Hereditary Endocrinopathies of the Federal State Budgetary Institution "National Medical Research Center of Endocrinology" of the Ministry of Health of Russia, Moscow. For DNA diagnostics, the author's panel of primers for multiplex polymerase chain reaction (PCR) and sequencing using the Ion Ampliseq ™ Custom DNA Panel technology (Life Technologies, USA) was used. The author's panel "osteogenesis imperfecta" included 13 genes whose products are involved in collagen modifications. Library preparation and emulsion PCR were performed according to the manufacturer's recommendations. Preparatory stage - preparation of DNA libraries: amplification of the studied regions of the genome, attaching adapters with 10 bar codes to them, purification of libraries; amplification of libraries on microparticles and enrichment of microparticles containing DNA templates. Next, the DNA sequence was deciphered (sequencing on a semiconductor sequencer Personal Genome Machine (Ion Torrent, Life Technologies, USA). Bioinformatic processing of sequencing results was carried out using the Torrent Suite 4.2.1 software module (Ion Torrent, Life Technologies, USA) and the Annovar software package (version 2014Nov12) [6].

The third stage

This analysis was carried out at the Medical Genetic Research Center (MGNTs), Moscow, using an ION S5 (Thermo Fisher Scientific, USA) device. Reading depth at least x110. This means that each investigated genome region is analyzed on average at least 110 times to avoid the influence of technical reading errors on the research results. For complex regions of the genome (for example, GC-rich regions), the average coverage may be lower.

The fourth stage

Molecular genetic diagnostics of patients was carried out in the laboratory of molecular pathology "Genomed", Moscow using MGISEQ-200 planform and kits (Beijing Genomics Institute, PRC). For molecular genetic analysis, the author's panel "Connective tissue diseases" was used. DNA analysis is carried out using a new generation sequencing technology using the pair-end reading method. The average coverage of the target sequencing sites in the studied genes is at least 70x.

Sequencing data processing is carried out using an automated algorithm that includes alignment of reads to the reference sequence of the human genome (hg19), post-processing of alignment, identification of variants and filtering of variants by quality.

Point 8

- Lines 91-92: “at the second stage, targeted NGS sequencing of 13 genes was carried out”. Please check the information as the figure 1 reports 18 genes in stage 2.

Response 8: In the figure 1 we made a mistake and we have fixed it in the new variant of article.

Figure 1. Four stages of research design

- As every stage of the analysis was carried out in different laboratories, how did the authors guarantee the storage, distribution and quality of the biological material collected?

DNA was extracted in our laboratory. After quality check DNA was sent to another laboratories in lyophilized state by licensed transport company. Before sequencing all necessary procedures of final quality check were done.

Point 9

Response 9

- Line 145: please correct the phrase to “OI database” as only one database was used as reference.

 We have changed the word " OI database " to the word " OI databases"

Point 10

Results

- Please harmonize the presentation of the data reported, it is confusing as it is now. I suggest to report the results as described in the methodology, divided into 4 stages.

Response 10

We accepted your suggestion and decided to add this part.

On the first stage we identified mutations in COL1A1 and COL1A2 genes that are in 80-90% are the cause of osteogenesis imperfecta. In COL1A1 gene: с.358C>T (p. Arg120X); с.375dupC (p. Ala126fs); с.407dupGG136fs; с.579delT (p.Gly194valfsX71) ; с.658C>T (p.Arg220X); с.858+1G>A; с.967G>T (p.Gly323X); с.1081C>T (p.Arg361X); с.1243C>T (p.Arg415Ter) ; с.2444delG (p.Gly815AlafsX293) ; с.2461G>A (p.Gly821Ser) ; с.2569G>T (p.Gly857Cys) ; с.2869C>T (p.Gln957X) ; с.3076C>T (p.Arg1026X); с.3792delG (p.Met1264fs); с.1354-12G>A. In COL1A2 gene: с.647G>A (p.Arg216His); с.874G>A (p.Gly292Ser); c.1197+5 G>A; с.1826G>A (p.Arg609Gln); с.1897_1902dupGCTGGT (p.Ala633_Gly634dup); с.2341G>C (p.Gly781Arg); с.2756G>A (p.Gly919Asp); с.2971G>C (p.Gly991Arg); с.3034G>A (p.Gly1012Ser); с.3277G>A (p.Gly1093Ser); с.3977A>G (p.Lys1326Arg).

On the second stage were found mutations in genes P3H1 (c.1051G>T, p.Glu351X) and IFITM5 (c.-14C>T).

On the third stage we found mutations in genes FGF23 (c.G212C, p.Ser71Thr), CLCN7 (c.141+4A>C) and TGFB1 (c.945G>C, p.Lys315Asn). 

On the fourth stage were identified mutations in LAMB3 (c.1903C>T, p.Arg635X), ALOX12B (c.526G>A, p.Glu176Lys), PLEKHM1 (c.2902-9C>T), ERCC4 (c.2395C>T, p.Arg799Trp), ARSB (c.454C>T, p.Arg152Trp), PTH1R (c.342C>A, p.His114Gln).

Point 11

- As explicit in the title all individuals included in the study are from the Republic of Bashkortostan, please avoid to repeat this information constantly as it is already implicit.

Response 11

We erased this information in the text.

Point 12

Conclusion

- I suggest to highlight in the conclusion that most of the mutations found by the authors are consistent with previously known OI main target genes (COL1A1 gene COL1A2 gene).

Response 12

We added sentense

In our research 93.1% of all identified mutations were found in I type of collagen genes (COL1A1/ COL1A2).

Point 13

Table 4

Patient 33. Did the patient had only short stature and 5 fractures during whole life as clinical manifestation? Why did he receive an OI type 4 diagnosis? Only short stature is not indication of OI.

Response 13

This patient had family history (father of patient №32) except clinical features that`s why he was included into investigation. We received new mutation after NGS and clinical manifestation was not typical for OI type 1-3.

Point 14

Patient 40. Did this patient have any feature of epidermolysis bullosa?

Response 14

No, she hasn`t any features of epidermolysis bullosa

Reviewer 2 Report

This paper reports on an approach to identify mutations for Osteogenesis imperfecta (OI) in patients from the Bashkortostan. In the sample of 62 patients from 52 families, by targeted resequencing, the authors found 16 mutations in COL1A1, 11 mutations in COL1A2, one mutation in P3H1 and IFIMT5 genes; 11 of them were not previously described in literature (3 in the COL1A1, 7 in COL1A2, and 1 mutation in the P3H1). They also found “combined mutations” in some patients with typical OI phenotypes.

This is a largely candidate-gene- resequencing study by a group of experts in rare-bone-disease genetics, who also performed in-silico validation of their findings. Most of the paper is descriptive though; it contains some hypothesis-generating information and confirmation of known data. The manuscript’s methodology is mostly up-to-date and gives evidence of the authors’ expertise in the field of rare-genetic-disease analysis. However, there are several minor concerns which require the authors’ attention:

“combined mutations” are mentioned in the Abstract & Table 4 but the term is not used in the text. Mediation by other genes is possible – but would not be fully captured by a targeted approach.

There is a discrepancy: at second stage, targeted NGS was done in 13 genes – but there are 18 in the Fig. 1.

Regarding a 5-years old with the ERCC4 mutation: cannot it be that OI is mis-classified? She indeed seems to have a Fanconi anemia (or can it be Cockayne Syndrome?)

  1. 56: fix “incomplete”; l. 123 & 142: check spelling; l. 136, 220: check wording;
  2. 486 has some grammatical inconsistency, pls. re-word.

In Discussion, pls. remind what you found while comparing with the others (l. 325 and 341-342).

Abbreviations:

NGS: a) is defined several times; b) “S” stands for sequencing – so cannot be followed by “sequencing” again. Define “HO”.

Scheme 1. First, why “Scheme” and not Table? Also, reword “described options”; consider replacing “First identified” with “novel” or “This study”.

Figure 1, legend: is  “structural” a correct word?

Author Response

Dear reviewer,

Thank You for attention to our manuscript and valuable recommendations. We accepted all your remarks and corrected text according to your recommendations.

Point 1: “combined mutations” are mentioned in the Abstract & Table 4 but the term is not used in the text. Mediation by other genes is possible – but would not be fully captured by a targeted approach.

Response 1: These mutations we have already mentioned in the lines 281-282 (Patient 33 had a combined mutation of c.2971G>C in the COL1A1 gene and c. 212G>C (p. Ser71Thr) in the FGF23 gene.)

The description of the patient 40 was accidentally dropped out of the text. We corrected and added it (The patient 40 had combined mutation of c.-14C>T in IFITM5 gene and c.1903C>T, p. Arg635X in LAMB3 gene).

Point 2:

There is a discrepancy: at second stage, targeted NGS was done in 13 genes – but there are 18 in the Fig. 1.

Response 2:

We corrected this item

Point 3

Regarding a 5-years old with the ERCC4 mutation: cannot it be that OI is mis-classified? She indeed seems to have a Fanconi anemia (or can it be Cockayne Syndrome?)

Response 3:

This patient actually had 2 diagnosis: an undifferetiated anemia and OI

Point 4

  1. 56: fix “incomplete”; l. 123 & 142: check spelling; l. 136, 220: check wording;
  2. 486 has some grammatical inconsistency, pls. re-word.

Response 4:

We corrected these points (56, 123, 142)

56-

All of them had clinical manifestations specific to of osteogenesis imperfecta: multiple fractures, blue sclera, deformities, deafness, dentinogenesis imperfecta, joint hypermobility and radiological data

123-

Molecular genetic diagnostics of patients was carried out in the laboratory of molecular pathology "Genomed"(Moscow), using MGISEQ-200 planform and kits (Beijing Genomics Institute, PRC).

142-

We deleted this sentence, because the next one repeats the previous sentence

136- We corrected this sentence

Annotation of the identified variants according to the canonical transcript 

of each gene and their prioritization was performed taking into account the ACMG recommendations version 3.0. 

220-We corrected this sentence

These mutations are not currently recognized as responsible for the development of OI result in alterations of the bone quality.

486 We corrected this sentence

Point 5

In Discussion, pls. remind what you found while comparing with the others (l. 325 and 341-342).

Response 5:

We changed sentences

325

In the Chinese population, structural changes account for 54%, and haploinsufficiency mutations account for 46% [11], in the Ukrainian population the ratio is exactly 49% / 51% [18], which differs from our sample of patients. We had structural mutations – 41%, and haploinsufficiency mutations account for 59%.

So, we compare percent haploinsufficiency mutations and structural changes of our populations with percent haploinsufficiency mutations and structural changes of other populations.

341-342

Mutations c.358C>T, c.658C>T, c.1243C>T, c.2869C>T, c. 3076C>T, c.858+1G>A, c.1354-12G>A, c.3208-1G>C were detected in patients with type I of OI. Patients from other populations had similar clinical manifestations to patients from our study. Mutations c.1081C>T, c.2461G>A and c.2569G>T found in COL1A1 gene lead to severe clinical symptoms in our patients with the type III of OI than in patients from literature who had type I of OI

Point 6

Abbreviations: NGS: a) is defined several times; b) “S” stands for sequencing – so cannot be followed by “sequencing” again. Define “НО”.

Response 6: We corrected these mistakes (a,b).

Point 7

Scheme 1. First, why “Scheme” and not Table? Also, reword “described options”; consider replacing “First identified” with “novel” or “This study”.

Response 7:

We have changed the word "Scheme" to the word "Table"

We have changed the words “described options” to the word “reference”

We have changed the word “First identified” to the- “novel”

Point 8

Figure 1, legend: is  “structural” a correct word?

Response 8: We have changed the name of the Figure 1 “Four stages of research design”
